# Comparing the Toxicological Responses of Pulmonary Air–Liquid Interface Models upon Exposure to Differentially Treated Carbon Fibers

**DOI:** 10.3390/ijms24031927

**Published:** 2023-01-18

**Authors:** Alexandra Friesen, Susanne Fritsch-Decker, Sonja Mülhopt, Caroline Quarz, Jonathan Mahl, Werner Baumann, Manuela Hauser, Manuela Wexler, Christoph Schlager, Bastian Gutmann, Tobias Krebs, Ann-Kathrin Goßmann, Frederik Weis, Matthias Hufnagel, Dieter Stapf, Andrea Hartwig, Carsten Weiss

**Affiliations:** 1Karlsruhe Institute of Technology (KIT), Institute of Applied Biosciences, Department of Food Chemistry and Toxicology, 76131 Karlsruhe, Germany; 2Karlsruhe Institute of Technology (KIT), Institute of Biological and Chemical Systems, Biological Information Processing, 76344 Eggenstein-Leopoldshafen, Germany; 3Karlsruhe Institute of Technology (KIT), Institute for Technical Chemistry, 76344 Eggenstein-Leopoldshafen, Germany; 4Vitrocell Systems GmbH, 79183 Waldkirch, Germany; 5Palas GmbH, 76229 Karlsruhe, Germany

**Keywords:** carbon fiber, pulmonary toxicity, air–liquid interface, co-culture, triple-culture, inflammation, gene expression, genotoxicity, apoptosis

## Abstract

In recent years, the use of carbon fibers (CFs) in various sectors of industry has been increasing. Despite the similarity of CF degradation products to other toxicologically relevant materials such as asbestos fibers and carbon nanotubes, a detailed toxicological evaluation of this class of material has yet to be performed. In this work, we exposed advanced air–liquid interface cell culture models of the human lung to CF. To simulate different stresses applied to CF throughout their life cycle, they were either mechanically (mCF) or thermo-mechanically pre-treated (tmCF). Different aspects of inhalation toxicity as well as their possible time-dependency were monitored. mCFs were found to induce a moderate inflammatory response, whereas tmCF elicited stronger inflammatory as well as apoptotic effects. Furthermore, thermal treatment changed the surface properties of the CF resulting in a presumed adhesion of the cells to the fiber fragments and subsequent cell loss. Triple-cultures encompassing epithelial, macrophage, and fibroblast cells stood out with an exceptionally high inflammatory response. Only a weak genotoxic effect was detected in the form of DNA strand breaks in mono- and co-cultures, with triple-cultures presenting a possible secondary genotoxicity. This work establishes CF fragments as a potentially harmful material and emphasizes the necessity of further toxicological assessment of existing and upcoming advanced CF-containing materials.

## 1. Introduction

There is a continuously rising demand for innovative and advanced materials in various sectors of industry. One material exhibiting several advantageous properties is carbon fiber (CF). It combines the characteristics of a low weight with a high tensile strength, a high Young’s modulus, as well as high thermal and chemical stability, which sets it apart from other materials such as steel or aluminum [1]. Two precursors are currently used for the synthesis of commercially available CF, polyacrylonitrile (PAN) and pitch, with PAN-based CF dominating the world market [2]. For the application as an advanced material, the virgin CFs (vCFs) are usually embedded in a polymeric matrix, yielding so-called carbon fiber reinforced polymers (CFRP). To enhance the interaction between CFs and the matrix, vCFs are typically coated with a sizing, changing the surface properties of the CF [3]. The aforementioned properties of CFs, and by extension CFRP, make them an important material for various lightweight applications in, e.g., the automotive, aerospace, or wind energy industries [4].

During the life cycle of CF, fiber fragments can be generated due to mechanical and thermal stress, starting at fiber production and further processing, through component manufacture, usage, recycling, and end-of-life recovery,. This includes fragments in the respirable range. As part of the CarboBreak project funded by the German Federal Ministry of Education and Research, high release levels of respirable fiber fragments in several mechanical processing operations, e.g., sawing and shredding of CFRP, were identified in workplace measurements and in a closed laboratory setup with inherent bulk sample analysis [5,6,7].

For recycling and other types of recovery of CFRP, thermal breakdown must be taken into account in addition to mechanical stress during preparation. It has been reported that at temperatures above 500 °C in an oxygen-containing atmosphere CFs undergo thermal decomposition, which takes place at the fiber surface and causes a reduction in fiber diameter [8]. Thus, respirable fiber fragments can be formed by an oxidation process, as the diameter of vCF is in the range of about 5 to 12 µm. In addition to a uniform decrease in fiber diameter, local oxidative fiber attack can take place, which results in uneven CF degradation and breaking points. This behavior must be considered especially when recovering end-of-life waste, which can diffusely enter incineration plants via household or commercial waste. In such plants, the CFs are not completely decomposed and can be discharged via residues [9].

The increasing use of CFs raises the question of their safety, both for working personnel in production or waste disposal and for consumers being in contact with CFRP during their life cycle. Similar to other fibrous materials, such as asbestos fibers or multi-walled carbon nanotubes (MWCNTs), CFs could exert possible toxicity through inhalation [10]. Fortunately, vCFs do not classify as respirable because of their large diameter (5–12 µm) and length. However, the different types of mechanical or thermal stress mentioned above can damage the fibers and result in the formation of smaller fragments in the respirable size range [11] being defined by the World Health Organization (WHO) as having a length l > 5 µm, a diameter d < 3 µm, and the aspect ratio l:d > 3:1 [12,13]. For respirable fibers, an increasing risk for diseases such as lung cancer is linked with an increasing rigidity and increasing biopersistence [14,15]. CFs are likely to have both those critical properties, being highly structured carbon materials [16]. Processes especially relevant for fiber fragmentation are the production and disposal of CF, with stress applied through, e.g., cutting or grinding and thermal processes [5,6,17,18].

Studies assessing the toxicity of CFs are scarce and have provided contradictory results: two investigations did not report any adverse effects of virgin PAN-based CF in rodents, while a specially designed virgin pitch-based CF with a smaller diameter of 1–4 µm, which is not representative for typical pitch-based CF, induced a transient inflammatory response [19,20]. Mechanically treated PAN-based CF samples containing mostly particles and less than 1% fiber-shaped fragments did not induce adverse effects in guinea pigs [21]. A more recent study compared the toxicological responses of mice exposed to virgin and thermally treated CF. Genes coding for proteins involved in DNA repair and chromatin remodeling were activated upon exposure to both materials [22]. Therefore, a comprehensive toxicological characterization of CF has not been performed yet. Comparing CF to other carbon-based materials such as carbon black particles (CB) and MWCNTs, CF can be assumed to be biopersistent [23]. Toxicological responses are likely to differ depending on the size and shape of the fiber fragments produced through mechanical or thermal treatment. Specifically, a fibrous shape with a high length-to-diameter ratio could promote various lung pathologies such as inflammation, fibrosis, and carcinogenesis [24].

This shape defines the pronounced toxicity of CNTs, which depends on various factors, such as the number of layers (multi-walled vs. single-walled), length, surface functionalization, and contamination with metals [25]. CNTs are known to possess a high inflammatory and toxic potential, which is characterized by cytokine secretion [26] and a subsequent induction of apoptosis, DNA damage, and possible tumor development [27]. The underlying mechanisms of the adverse effects promoted by CNTs and other carbon-based materials have been found to rely on the generation of reactive oxygen species (ROS) and lysosomal dysfunction [25]. Still, CNTs provoke a stronger response than the equally carbon-based but mostly spherical CB, which are classified as a material with poor solubility and low toxicity (PSLT) [28]. Consequently, many studies have reported CB to only be toxic under overload conditions, which do not reflect a realistic exposure scenario [29]. Nevertheless, CB are also known to induce inflammation and ROS formation [30,31], which is put into context with a possible mechanism of secondary genotoxicity [32].

In recent years, several advanced models have been presented to assess the pulmonary toxicity of particles and fibers in vitro. This development occurred in the context of the 3 Rs (replace, reduce, and refine) concept, to ensure a reduction in in vivo testing both for ethical reasons as well as better interspecies comparability [33]. In light of this, models culturing lung epithelial cells at an air–liquid interface (ALI) have become the preferred method to replicate physiological conditions in the lung more closely. ALI-grown cells are cultured on a porous membrane, being supplied with a medium from the basolateral compartment and exposed to ambient air or aerosol from the apical side [34,35]. Several studies assessing particle and fiber toxicity in ALI-grown cells have been performed, e.g., with wood [36,37] or cigarette smoke [38], CNTs [39], or quartz particles [40,41]. Furthermore, epithelial mono-cultures can be complemented with other cell types such as macrophages [42,43] or fibroblast cells [44,45] to account for interactions between different cell types in the lung.

In the present study, it was therefore our aim to investigate the toxicity of CF samples in advanced in vitro models of the human lung. To this end, three different models were applied: a mono-culture model consisting of BEAS-2B cells (bronchial epithelia); a co-culture model further containing differentiated THP-1 cells (dTHP-1: macrophage-like cells); and a triple-culture model with BEAS-2B, dTHP-1 cells in the apical compartment, and CCD-33Lu cells (fibroblasts) in the basolateral compartment. A schematic representation of the three models can be viewed in Appendix A. A PAN-based CF is either treated mechanically in a planetary ball mill or thermo-mechanically, first in two furnaces and subsequently in a planetary ball mill, with the resulting samples being fed into an exposure station containing the different cell models. The toxicity of the differentially treated CF is assessed by the endpoints of cytotoxicity, transcriptional profiling of toxicity markers, pro-inflammatory responses, and genotoxicity.

## 2. Results

Three different cell culture models, BEAS-2B mono-, BEAS-2B/dTHP-1 co-, and BEAS-2B/dTHP-1/CCD-33Lu triple-culture systems, were exposed to CFs that underwent prior mechanical or thermo-mechanical treatment. While the mono- and co-culture systems were exposed to mechanically (mCF) and thermo-mechanically treated CF (tmCF) and post-incubated for 0, 3, or 23 h, the triple-cultures were exposed to mCF only and post-incubated for 23 h because of the complexity of the system. For all experiments, cells were exposed to one dose of pre-treated CF, which fluctuated between different experiments and thus is given as a mean value with standard deviation. The deposited dose was monitored online by a quartz crystal microbalance (QCM) and additionally determined by analysis of digital microscopy images. The focus was set specifically on the time-course analysis, since previous research has shown a great influence of the chosen time points on noxious responses, especially with respect to inflammation [41,46]. Following post-incubation, cells and cell culture medium were harvested for toxicological assessment. All results are shown comparing CF-exposed samples and their corresponding clean air controls (CACs). Firstly, cytotoxicity was evaluated by the lactate dehydrogenase (LDH) assay and the determination of the cell count. Furthermore, gene expression profiles of the cell culture models were generated by a high-throughput RT-qPCR method used in recent studies [47]. Only changes above a log_2_-fold value of +1 or below −1 were considered biologically relevant, which corresponds to doubling or halving of the transcript levels [48,49,50,51]. To further assess the inflammatory potential of CF, interleukin-8 (IL-8) ELISAs were conducted. Finally, the generation of DNA strand breaks was assessed via the Alkaline Unwinding method.

### 2.1. Physico-Chemical Properties of Deposited Carbon Fibers

The investigated CF was a high modulus fiber with mechanical characteristics being a tensile strength of 4700 MPa, a Young’s modulus of 390 GPa, and a nominal fiber diameter of 4.9 µm. The polyacrylonitrile (PAN)-based CF had a carbon content >95% and a density of 1.78 g/mL, a melting point of around 3500 °C and a degradation temperature higher than 650 °C in air as specified by the manufacturer. The specified sizing is composed of poly-(bisphenol A epichlorohydrin), and CF may contain resins with an average molecular weight of ≤700 g/mol formed as a reaction product (0.1 to <1%). The morphological characteristics of the treated and untreated CF are depicted in Figure 1. After treatment of virgin CF, the percentage of fragmented fibers and particles increased.

In Table 1, the exposure and deposited dose, as well as the relative percentage of fibers and particles, are shown.

### 2.2. Cytotoxicity upon Exposure to Pre-Treated Carbon Fibers

Cytotoxicity was assessed via the LDH assay and determination of cell count. The results for the mono- and co-cultures are depicted in Figure 2; the results for the triple-cultures can be viewed in Figure 3.

Exposure to mCF did not elicit significant cytotoxic reactions in both culture systems. LDH release ranged from 3 to 9% in CACs and CF-exposed cells. The cell count rose from 8.5–9.5 × 10^5^ cells after 1 h of exposure and 0 h of post-incubation to 12.5–13.5 × 10^5^ after 23 h of post-incubation. Similarly, for tmCF, no significant differences were apparent in the LDH assay, with LDH release ranging from 2 to 20% in CACs and CF-exposed cells. In contrast, this fiber treatment resulted in a reduction in the cell count. While CACs continued to proliferate, the counts of CF-exposed cells dropped to 4.5–6.0 × 10^5^ cells in the mono-culture and 6.0–7.0 × 10^5^ cells in the co-culture.

No difference in the LDH release of triple-cultures exposed to mCF was detected. Moreover, in the apical compartment containing BEAS-2B and dTHP-1 cells, no significant changes in cell count occurred. Contrastingly, the CCD-33Lu cells in the basolateral compartment displayed a small but significant reduction in cell count, from 2.27 × 10^5^ cells in the CACs to 1.95 × 10^5^ cells in the CF-treated samples.

### 2.3. Transcriptional Toxicity Profiles upon Exposure to Pre-Treated Carbon Fibers

#### 2.3.1. BEAS-2B Mono- and BEAS-2B/dTHP-1 Co-Cultures

Gene expression profiles of mono- and co-cultures after exposure to mCF and tmCF were generated. The results are depicted as a heatmap in Figure 4. Only genes with changes above a log_2_ value of +1 or below −1 are shown. The results of the whole gene set can be viewed in Appendix A.

For mCF, the most apparent changes occurred in the cluster “inflammation”. In the mono-culture, genes such as *IL-6* and *IL-8* showed a clear 3.2-fold and 2.8-fold upregulation, respectively, immediately after exposure (1 h exposure and 0 h post-exposure; 1 + 0), returning to baseline levels 23 h later. The co-culture exhibited the strongest upregulation of 4.6-fold for *IL-6* and 2.6-fold for *IL-8* (1 + 0 h), as well and a slower decrease in upregulation over time. No biologically relevant changes below a log_2_ of −1 or above +1 were observed in the clusters “oxidative stress response” and “apoptosis and cell cycle regulation”. However, the genes *DDIT3* and *GADD45A* from the cluster “DNA damage response and repair” showed a slight induction in both culture systems.

In contrast, exposure to tmCF resulted in stronger transcriptional changes. Similar to the results described above, *IL-6* and *IL-8* showed the strongest activation patterns in the “inflammation” cluster. In the mono-culture, expression of these two genes peaked at a time point of 1 + 3 h with an 18- and 7.8-fold upregulation, respectively. Contrarily, the co-culture exhibited a lower upregulation at that time point with 5.0- and 2.3-fold, respectively. Interestingly, *CCL22* and *TNF-A* were downregulated in the co-culture at all time points, with a maximum 2.8- and 3.5-fold downregulation after 1 + 23 h. *COX2* and *IL-1B* were repressed after 1 + 23 h as well. The cluster “oxidative stress response” showed a downregulation of *HMOX1* and an induction of *HSPA1A* and *NFKBIA.* The genes *JUN*, *MYC*, *PLK3*, *PMAIP1*, *TNFSRF10B*, and *VEGFA*, which are involved in apoptotic signaling, were also affected, with most of these genes displaying the highest induction after 1 + 3 h in both culture systems. Repeatedly, the induction was slightly higher in the mono-culture. Similar but more pronounced than the results observed for mCF, exposure to tmCF resulted in the induction of DNA damage markers *DDIT3* and *GADD45A. DDIT3*, together with *IL-6*, was marked as one of the genes influenced most strongly, with a 19-fold induction in the mono-culture after 1 + 3 h.

#### 2.3.2. BEAS-2B/dTHP-1/CCD-33Lu Triple-Cultures

Transcriptional changes upon exposure with mCF at a post-incubation time of 23 h were assessed in the triple-culture as well. The results are depicted in Figure 5 as a heatmap. Only results for the apical compartment containing BEAS-2B and dTHP-1 cells are displayed. The results of the whole gene set can be viewed in Appendix A.

The inflammatory gene expression profile of the triple-culture stands in contrast to the profiles of the mono- and co-cultures. A repression was noted for most of the genes, especially for *COX2* (2.6-fold) and *TNF-A* (2.9-fold). The fibrotic gene *OPN* displayed a relevant 3.2-fold downregulation as well. Apart from these genes, no relevant changes beyond the threshold log_2_ value of +1 or −1 were observed, which reflects the effects in mono- and co-cultures after 1 + 23 h.

### 2.4. Cytokine Release upon Exposure to Pre-Treated Carbon Fibers

Since a strong inflammatory response had been observed on the transcriptional level, the release of inflammatory proteins was assessed by IL-8 ELISA. The results for mono- and co-cultures are displayed in Figure 6, while the IL-8 release of triple-cultures is shown in Figure 7. Cells incubated with 10 µg/mL lipopolysaccharide (LPS) were used as a positive control.

The IL-8 release mirrored the corresponding expression of the *IL-8* gene. Exposure to mCF led to a significant IL-8 release in the co-culture, peaking at 354 pg/mL after 1 + 3 h, while the mono-culture released less IL-8 at all time points. Similarly, more IL-8 was released after exposure of the co-culture to LPS (6.96 ng/mL) than after exposure of the mono-culture (1.28 ng/mL), with the release being at least 14 times higher than after exposure to CF altogether. The highest IL-8 release appeared at a later time point than the maximum upregulation in the corresponding gene expression: the transcriptional activation of *IL-8* peaked at 1 + 0 h for both culture systems, while the release of the protein occurred after 1 + 23 h in the mono- and after 1 + 3 h in the co-culture system.

tmCF triggered a higher IL-8 release than mCF in both culture systems. Few differences were observed between them, with mono- and co-cultures releasing a maximum of 773 pg/mL and 563 pg/mL, respectively. Again, a time shift between transcriptional and secretory response was observed: the expression of *IL-8* peaked after 1 + 3 h; the secretion of the corresponding protein, after 1 + 23 h.

The triple-culture released an extraordinary amount of IL-8 after 1 + 23 h, with a release of 4.92 ng/mL after exposure to mCF and a release of 19.9 ng/mL after exposure to LPS. These values exceeded the IL-8 release in mono- and co-cultures by a factor of at least 13.8 and 2.8 for CF and LPS, respectively.

### 2.5. Induction of DNA Strand Breaks upon Exposure to Pre-Treated Carbon Fibers

The transcriptional profiles indicated a possible involvement of DNA damage signaling and repair in the toxicological response to CF via the induction of the two genes *DDIT3* and *GADD45A*. To determine whether CFs really trigger genotoxicity, the formation of DNA strand breaks after exposure to mCF and tmCF was assessed. To this end, the Alkaline Unwinding method was applied. With this method, the formation of DNA single-strand breaks is indirectly monitored by a reduced amount of double-stranded DNA. The results for mono- and co-cultures are depicted in Figure 8. The effect of mCF on triple-cultures is depicted in Figure 9.

Exposure to mCF resulted in a slight but significant reduction in double-stranded DNA after 1 + 3 h in mono- and co-cultures. Similarly, tmCF caused the same effect, which was also apparent after 1 + 23 h in the co-culture. Thus, no or merely a slight induction of DNA strand breaks was observed.

In contrast, exposure to the triple-culture resulted in a clear and significant reduction in double-stranded DNA. Whereas about 62% of double-stranded DNA was present in untreated cells, in CF-exposed cells this amount was reduced to 44%. Therefore, a moderate induction of DNA strand breaks was induced in the triple-culture.

## 3. Discussion

In this work, we aimed to assess the toxicity of carbon fiber (CF), a material that has not been investigated in great detail with regard to toxicity yet. Most CFs exceed the respirable size range both in diameter and length. However, to mimic stress scenarios encountered in the life cycle of CF such as pyrolysis and shredding with relevance for recycling and disposal, the investigated PAN-based CF was pre-treated to achieve smaller fiber fragments estimated to be able to enter the respiratory tract [11]. In this context, the CFs underwent either mechanical treatment in a planetary ball mill (mCF) or thermal treatment in a tube reactor followed by mechanical treatment in a planetary ball mill (tmCF). The resulting samples were injected into the Vitrocell^®^ Automated Exposure Station and deposited onto mono-, co-, and triple-cultured cells reflecting the physiology of the lung in varying degrees of complexity. After one hour of exposure, the cultures were post-incubated for 0, 3, or 23 h. Afterwards, the toxicological responses were evaluated considering the endpoints of cytotoxicity, transcriptional toxicity, cytokine secretion, and genotoxicity.

The CF fragments deposited on the surfaces were a mixture of particles, fibers, and WHO fibers with a significant fraction of WHO fibers (Table 1). Compared to the mCF, for the tmCF, a lower fraction of fibers and WHO fibers was observed, whereas the particle fraction increased. On the other hand, the aerodynamic equivalent diameter d_ae_ of the deposited tmCF (4.9 ± 0.4 µm) was larger than the d_ae_ of the deposited mCF (3.7 ± 0.9 µm).

Exposure to mCF did not induce cytotoxic effects in mono- or co-cultures, whereas a significant albeit only slight reduction in the cell count was observed for the fibroblasts in the triple-culture. Exposure to tmCF also did not enhance LDH release, but reduced cell counts significantly in both mono- and co-cultures at all time points. A lower cell count by the reduced proliferation of the cells can be ruled out, since more cells had been seeded than harvested at the end of the experiments. Cell loss without corresponding cellular damage might be explained by the strong agglomeration of the CF fragments, which was observed during cell harvesting. Presumably, the sizing coating of the fibers is removed by the thermal treatment, which exposes the bare carbon surface underneath. This results in a strong agglomeration and subsequent entrapment of cells inside the agglomerates, which is in line with the reduced cell counts upon exposure to tmCF. Strong agglomeration of carbon-based materials has already been described for carbon black (CB) particles [52,53], as well as an affinity of CB to dipalmitoylphosphatidylcholine, a major constituent of biological membranes and lung surfactant [54]. The low cytotoxicity is in line with the few other studies conducted with CF, with vCF inducing a low to moderate cytotoxic response in rabbit macrophages [55] and no cytotoxicity in rats upon inhalation [19]. Thermally treated composites comprised of graphene oxide and CF elicited a moderate cytotoxic response at a high concentration of 120 µg/mL, which can be considered overload conditions [56]. Thus, both vCF, mCF, and tmCF do not provoke classic cytotoxicity. Other carbon-based materials are known to induce cytotoxic effects depending on their shape; while fibrous materials, such as MWCNTs, are reported to be cytotoxic in rat macrophages [57], BEAS-2B cells [58], and A549/dTHP-1 co-cultures [59,60] at doses below 10 µg/cm^2^, particulate materials such as CB show no [52] or low cytotoxicity in A549 cells [61]. In conclusion, the absent cytotoxicity of mCF or tmCF appears to resemble the low cytotoxicity of CB particles and differs from the more pronounced effects induced by CNTs.

Gene expression analysis unveiled further differences in the toxic responses to the differentially treated CF. Generally, exposure to mCF elicited moderate responses, the most prominent changes being present in the inflammation cluster for the genes *IL-6* and *IL-8*. Both peaked immediately after exposure, with the co-culture showing a persistent induction at later time points as well. This was further reflected in the release of IL-8, which confirmed that the co-culture was more active than the mono-culture evident by the secretion of higher amounts of IL-8. Exposure to tmCF triggered stronger effects, especially in BEAS-2B mono-cultures. Enhanced expression of *IL-6* and *IL-8* appeared similar in both cell culture models immediately after exposure, and peaked after three more hours of post-incubation. Cytokine release mirrored these results, with a higher overall release of IL-8 and no significant differences between mono- and co-cultures.

Another observation in the co-culture was the repression of *CCL22* and *TNF-A*, two genes coding for the macrophage-derived chemokine (MDC) and tumor necrosis factor α (TNF-α), respectively. Both proteins are key regulators of macrophage activity [62,63]. This effect, however, might be attributed at least in part to the artificial phenomenon of cell loss due to adhesion to thermally treated CF discussed earlier, which could preferentially deplete the dTHP-1 cells as they come into contact with the CF fragments first. To what extent this artifact might contribute in general to the overall lower transcriptional response of the co-culture as well as the cytokine release warrants further investigations.

Regarding intracellular signaling, the involvement of *TNF-A*, *IL-8*, and *IL-6*, as well as *NFKBIA*, coding for the NF-κB inhibitor α, suggests an activation of the NF-κB pathway [64]. Moderate inflammasome activation is also suggested, given the induction of *IL-1A* and *IL-1B*, coding for the cytokines IL-1α and IL-1β [65]. For tmCF, the response does not seem to rely on oxidative stress primarily, as only a few genes of this cluster were affected at all, with *HMOX1* coding for the heme oxygenase 1 (HO-1) even being repressed. This could be related to the repression of nuclear factor erythroid 2-related factor 2 (Nrf2) transcription factor activity; although, the underlying mechanism still needs to be elucidated [66]. The alteration of genes related to apoptosis and cell cycle regulation indicates a general stress response. As such, the gene *JUN* encodes one subunit of the proapoptotic dimeric transcription factor activator protein (AP-1) [67,68], which is activated for example by TNF-α via the c-Jun N-terminal kinase (JNK) pathway [69]. Furthermore, the gene *DDIT3* showed the highest activation in the whole gene set. *DDIT3* encodes the DNA damage-inducible transcript 3, which, apart from its role in DNA damage, is closely related to proapoptotic signaling. It promotes apoptosis via the intrinsic pathway through the repression of Bcl2 (B-cell lymphoma 2) [70] as well as via the extrinsic pathway via death receptor 5, the activation of which is suggested by the upregulation of its coding gene, *TNFRSF10B* [71]. At the same time, the activation of some marker genes related to antiapoptotic signaling was observed through the induction of *VEGFA*, encoding the vascular endothelial growth factor α (VEGF-α), which can induce the transcription factor c-Myc (represented by the corresponding gene *MYC*) via the MAPK/ERK pathway [72,73]. Therefore, both pro- and antiapoptotic signaling seem to be affected by exposure to tmCF. Induction of the two genes *PMAIP1* and *PLK3*, coding for the respective proteins Noxa and Polo-like kinase 3 (Plk3), further alludes to the involvement of the tumor suppressor p53. In this context, Plk3 provides a connection between DNA damage signaling and p53 activation [74], whereas Noxa promotes apoptosis via the inhibition of antiapoptotic Bcl-2 proteins and is directly induced by p53 [75]. On the other hand, other genes known to be induced by p53, such as *MDM2*, appeared unaffected by exposure to tmCF, and therefore only a subset of p53 target genes are regulated [76].

An important role of TNF-α release has been described for CF-exposed murine macrophages [56], which is unfortunately the only study assessing inflammatory or other metabolic changes after CF exposure. Other carbon-based materials, such as CB, MWCNTs, or graphene oxide plates, are known to induce strong inflammatory responses in vivo in rats [77] as well as in vitro in murine macrophages [57,78,79]. The toxicity and inflammation provoked by carbon-based nanomaterials are generally attributed to a generation of ROS and lysosomal dysfunction, which can result in the activation of transcription factors such as NF-κB and AP-1 as well as inflammasome activation [25]. These findings are in line with the effects observed in the present study, those reported for MWCNTs in A549/dTHP-1 co-cultures at the ALI triggering release of IL-8 [59] and various other studies in submerged lung cell models [80,81,82]. Contrastingly, carbon nanoparticles have been found to induce the gene *HMOX1* significantly in 16HBE14o- cells at the ALI, which poses a contrast to the present data [83]. Nevertheless, apoptotic pathways have been found to be induced by various carbon-based substances through the activation of the JNK, AP-1, and MAPK/ERK pathways, for instance in vivo [84] or in primary rat epithelial cells [85]. For CB particles in particular, the onset of pyroptosis, a highly inflammatory type of programmed cell death, has been described [86]. This mechanism of cell death can be ruled out in the present work, as pyroptosis also involves a strong LDH release via pore formation [87].

For the triple-culture experiments, no biologically relevant changes at the level of gene expression were observed, with the exception of a slight downregulation of *COX2*, *TNF-A*, and *OPN*, which are all involved in inflammatory or fibrotic pathways. As in mono- and co-cultures, the most significant up- or downregulation occurred at earlier time points, the selected time point of 23 h post-exposure for the triple-culture was presumably too late to capture changes in gene expression. This assumption is reinforced by the exceptionally high IL-8 release noted in the ELISA. Another possibility is the occurrence of a type 2 inflammatory response, which would be characterized by an induction of anti-inflammatory cytokines such as IL-4 or IL-13, which were not covered by the gene set. However, type 2 inflammation would also be visible through the induction of fibrotic genes, such as *TGF-B*, *PDGF-A*, or *TIMP1*, which was not observed in this work [88]. Generally, the integration of multiple cell types, especially epithelial cells, macrophages, and fibroblasts, in studies of carbon-based materials, has been reported to result in strong inflammatory and fibrotic changes, both in conditioned media approaches [89], organized 3D microtissues [90], and the commercially available EpiAlveolar model [91]. Most of these changes have been found to occur after repeated application of the materials over the course of a few days or weeks. Therefore, the chosen time point for triple-culture studies analyzing gene expression should include early time points, i.e., a few hours after exposure, but also prolonged and repeated exposure needs to be considered.

Lastly, genotoxicity was assessed with the alkaline unwinding assay. A slight, but significant induction of DNA strand breaks was observed upon deposition of both mCF and tmCF after 1 + 3 h. Indeed, gene expression analysis revealed an activation of the genes *DDIT3* and *GADD45A* from the DNA damage cluster, both of which, however, can also be induced by various other cellular stress reactions, such as ER stress or apoptosis [92,93]. This suggests a modest genotoxic response. In contrast, MWCNTs are reported to be genotoxic based on the formation of micronuclei [60], DNA strand breaks [94], and fragmentation and translocation of chromosomes [95]. The pronounced genotoxicity of the fiber-shaped MWCNTs is generally attributed to a high degree of frustrated phagocytosis and consecutive formation of ROS [96]. In the case of CF exposure, not only are fiber-shaped objects deposited but a large fraction of particles as well. Thus, mechanisms leading to genotoxicity initiated by processed CF are more comparable to those observed for CB particles, which are characterized by inflammation-driven pathways of secondary DNA damage [32] and are mostly known to occur under overload conditions [97]. Incidentally, for the triple-culture, a more robust formation of DNA strand breaks was observed in the apical compartment containing BEAS-2B and dTHP-1 cells. Since this effect was not observed in the BEAS-2B/dTHP-1 co-culture without fibroblasts, a mechanism of secondary genotoxicity due to the mutual interaction of fibroblasts with the epithelial cells and macrophages is suggested. The high IL-8 release observed in the triple-culture could be one of the driving forces leading to such a secondary genotoxicity as observed here. Previously, in a co-culture model of human trophoblast-derived BeWo cells and fibroblasts, an indirect effect across cellular barriers promoting DNA damage could be shown. In this case, nanoparticle-exposed BeWo cells released soluble ATP, which induced genotoxicity in unexposed fibroblasts via receptor-mediated signaling [98].

To summarize our findings, the two different treatments of PAN-based CF were found to induce distinct toxicological responses upon exposure to three cellular models mimicking the human lung. Mechanical treatment resulted in a visible fragmentation of the fibers and a moderate toxicological response, characterized primarily by pro-inflammatory reactions. Contrastingly, thermo-mechanical treatment produced a higher amount of small fiber fragments and resulted in a desizing of the fibers and fiber fragments. This change in surface chemistry led to a stronger cellular response, characterized by a pro-inflammatory and general stress response.

In the current manuscript, we only examined one specific CF subjected to two different types of treatment. Clearly, this first and thus limited study needs to be expanded given the importance of this topic. Future work should focus on a broader toxicological characterization of PAN- but also pitch-based CF. In particular, different fiber treatments need to be considered that try to mimic realistic scenarios reflecting different stages of the CF life cycle with particular emphasis on the release of WHO fibers. Such toxicological investigations should be complemented by exposure assessment, e.g., at different workplaces, and surveillance of potential adverse health effects.

## 4. Materials and Methods

### 4.1. Materials

All chemicals, media, and supplements for cell culture and ALI exposure were purchased from Gibco/Thermo Fisher Scientific GmBH (Dreieich, Germany) and Roth (Karlsruhe, Germany), except for KGM, which was acquired from Lonza (Basel, Switzerland). The PAN-based CF Tenax^®^-J UMS40 was purchased from Teijin Carbon Europe GmbH (Wuppertal, Germany). Cell culture consumables were obtained from Greiner (Kremsmünster, Germany), and Transwell^®^ plates and inserts were produced and purchased from Corning (Amsterdam, The Netherlands). LDH assay kits were bought from Roche (Mannheim, Germany). Triton X-100 was purchased from Roth (Karlsruhe, Germany). Primers for RT-qPCR were synthesized by Eurofins (Ebersberg, Germany). PCR reagents were acquired from Macherey-Nagel (Dueren, Germany), Applied Biosystems (Foster City, CA, USA), Teknova (Hollister, OH, USA), Fluidigm (San Francisco, CA, USA), Bio-Rad (Munich, Germany), and New England Biolabs (Frankfurt am Main, Germany). IL-8 ELISA kits were obtained from Invitrogen (Waltham, MA. USA). LPS and Menadione were purchased from Sigma-Aldrich (Steinheim, Germany).

### 4.2. Fiber Preparation and Characterization

Two pre-treatment routines were developed to achieve a CF bulk material suitable for aerosolization with a segmented belt aerosol generator. In the case of mechanical stress, the CFs were cut by hand into pieces of 1 cm and milled for 8 min at 400 rpm in a planetary ball mill (Retsch Planetary Ball Mill PM 100 CM, Retsch GmbH., Haan, Germany). The thermo-mechanically treated CF was also first cut by hand into 1 cm pieces, but subsequently subjected to a two-step thermal treatment before milling at 500 rpm for 2 min. First, the CF pieces were desized in a muffle furnace for four hours (Thermoconcept, Model HTL 04/17, Nabertherm GmbH, Lilienthal, Germany) at 400 °C in 8 L/min nitrogen. Further thermal stress was induced in a continuous flow fixed bed. The fixed bed was heated up to 800 °C in a nitrogen atmosphere inside the quartz tube reactor of a tube furnace (TS split tube furnace TS, Carbolite Gero, Neuhausen, Germany). After heating up, the gas was switched to 1 L/min air for 30 min. Cooling down took place in nitrogen atmosphere as well. The mCF and tmCF samples were characterized by image analysis of digital micrographs taken with a VHX-6000 (Keyence Deutschland GmbH, Neu-Isenburg, Germany).

For aerosol generation, the mCF and tmCF samples were fed into the container of a segmented belt aerosol generator (Dust Aerosol Generator 3410, TSI GmbH Aachen, Germany) and dispersed with dry air. The aerosol passed a retention reactor and was directed into the aerosol exposure station via stainless-steel tubing. All parts of the setup were grounded to minimize losses due to electrostatic forces (Appendix A).

During exposure experiments, the aerosol inside the exposure station was characterized for number concentration using an optical particle counter (welas promo 2000, Palas GmbH, Karlsruhe, Germany). The surface dose applied in the exposure chambers was determined online by QCM [99] as well as by image analysis of the deposited particles and fibers on the surface of the quartz crystal sensor.

All micrographs were analyzed in a two-step method: generation of a binary image, object detection, and measurements of the object geometries were performed with FibreShape (IST AG, Vilters, Switzerland). In a further step, the object classifications were statistically evaluated regarding different morphological criteria such as shape (particle or fiber) and the aspect ratio (i.e., the ratio of length to diameter), with the WHO criteria for respirable fibers being a minimum length of 5 µm and a maximum diameter of 3 µm. An example of the single object analysis from a tmCF sample concerning length and diameter with classification of objects such as particles, fibers, and WHO fibers can be seen in Appendix A.

### 4.3. Cell Culture

The human bronchial epithelial cell line BEAS-2B was obtained from American Type Culture Collection (ATCC^®^ CRL-9609) [100]. BEAS-2B cells were grown in Keratinocyte Growth Medium (KGM), supplemented with all additives provided by the manufacturer as part of the KGM-2 Bullet Kit. Prior to sub-cultivation or experiments, all growth surfaces were coated with 10 µg/mL fibronectin, 30 µg/mL rat-tail collagen, and 10 µg/mL bovine serum albumin. The coating mixture was removed after 30 min and surfaces were washed with phosphate-buffered saline (PBS). The human monocytic cell line THP-1 was purchased from DSMZ (Braunschweig, Germany) and cultivated in RPMI-1640 medium, supplemented with 10% fetal bovine serum (FBS), 100 U/mL penicillin, 100 µg/mL streptomycin, and 1% L-glutamine at 37 °C with 5% CO_2_ [101]. The monocytic cell line THP-1 can be differentiated to achieve a macrophage-like phenotype by incubation with phorbol-12-myristate 13-acetate (PMA) [102,103]. For ALI exposure, THP-1 cells were differentiated (dTHP-1) by incubation with 30 ng/mL PMA. After five days, the medium was removed and the adherent cells were cultured in PMA-free medium for an additional three to five days. The human pulmonary primary fibroblast cell line CCD-33Lu was purchased from ATCC (CRL-1490). CCD-33Lu cells were cultivated in DMEM-HG medium, supplemented with 10% FBS, 100 U/mL penicillin, 100 µg/mL streptomycin, and 1% L-glutamine. All three cell lines were kept at 37 °C in a humidified atmosphere of 5% CO_2_ and detached using Accutase^®^. Only passages 40–60, 3–30, and 3–7 were applied in cell culture experiments for BEAS-2B, THP-1, and CCD-33Lu cells, respectively.

### 4.4. Exposure via the Vitrocell^®^ Automated Exposure Station (AES)

#### 4.4.1. Preparation of Cell Culture Models for ALI Exposure

For the mono-culture model, 6-well Transwell^®^ insert membranes with a surface area of 4.67 cm^2^ and a pore size of 0.4 µm were coated as described previously and BEAS-2B cells were seeded into the apical compartment at a density of 2 × 10^5^/cm^2^. For the co-culture model, BEAS-2B cells were seeded in the same manner, followed by seeding of dTHP-1 cells into the apical compartment at a density of 3.9 × 10^4^/cm^2^ after four hours. For the triple-culture model, the inserts were inverted and placed into sterile Petri dishes after the coating procedure. Then, 1 mL of medium containing 4.2 × 10^5^ CCD-33Lu cells was carefully transferred onto the bottom of the Transwell^®^ membrane. Within one hour, the fibroblasts settled down and attached to the membrane surface. Then, the inserts were inverted and placed into 6-well plates filled with pre-warmed medium. After an additional hour in the incubator, BEAS-2B and dTHP-1 cells were seeded as described for the co-culture model. The cultures were left in the incubator overnight under submerged conditions. All cells were seeded in RPMI-1640 medium, supplemented with 10% FBS, 1% penicillin/streptomycin, and 1% L-glutamine.

#### 4.4.2. Exposure via Vitrocell^®^ AES

A day prior to exposure, the AES was cleaned and prepared for the exposure procedure. For ALI exposures, the apical medium was carefully removed from the cell cultures, while the basolateral medium was replaced with fresh medium suited for the exposure of cells: RPMI-1640, supplemented with 4-(2-hydroxyethyl)-1-piperazineethanesulfonic acid (HEPES), 1% penicillin/streptomycin, and 1% L-glutamine. To avoid a decrease in temperature, the plates containing the inserts were then transferred into an isolated box and transported to the AES. The wells of the exposure modules were filled with 6.3 mL of pre-warmed exposure medium and the inserts were placed into the exposure wells. Afterward, the modules were closed and the exposure was started.

The exposure took place in the Vitrocell Automated Exposure Station (AES), which is set up for CF exposure according to previous experience with nanoparticles, with a temperature of 37 °C, relative humidity of 85%, and flow rate of 100 mL/min [36,104,105,106]. For use with CFs, the size-selective inlet was changed from PM_2.5_ to PM_10_ as the aerodynamic equivalent diameter of fibers with a diameter d = 3 µm and a length l = 10 µm is calculated to d_ae_ = 5.62 µm according to [107].

After one hour of exposure, the inserts were transferred back into 6-well plates and transported back to the biological laboratory. The exposure medium at the basolateral compartment was replaced by post-incubation medium (RPMI-1640, supplemented with 1% penicillin/streptomycin, and 1% L-glutamine). The cell cultures were post-incubated at 37 °C in a humidified atmosphere of 5% CO_2_ for 0, 3, or 23 h, respectively.

#### 4.4.3. Harvesting of Cells

After the required post-incubation time elapsed, medium was carefully removed from the cell cultures and transferred to 15 mL centrifuge tubes for cytokine detection and LDH release. To detach the cells from the membrane, 0.05% trypsin-EDTA solution or accutase^®^ was added to the apical and basolateral compartments. After an incubation time of 5 min, cell culture medium was added and cell suspensions were transferred to Eppendorf tubes. Cells were counted using a hemocytometer. Then, 3 × 10^5^ cells were transferred to a new tube, centrifuged (1300 rpm, 4 °C, 4 min), and 40 µL of phosphate-buffered saline (PBS) was added to achieve a cell suspension suitable for Alkaline Unwinding. The rest of the cells were centrifuged (1300 rpm, 4 °C, 4 min), washed with PBS, and the resulting pellets were frozen until gene expression analysis was performed.

### 4.5. Cytotoxicity Assessment via Lactate Dehydrogenase (LDH) Assay

Lactate dehydrogenase (LDH) release was investigated using the Cytotoxicity Detection Kit (Roche, Mannheim, Germany) following the manufacturer’s instructions. Briefly, medium supernatant was transferred into a 96-well plate and incubated with the same amount of catalyst/dye solution. After 5–10 min of incubation time in the dark, 1 N HCl was added to stop the reaction. Absorbance at 490 nm was measured using a plate reader. Positive control cells were treated with 0.1% Triton X-100 for 30 min before the end of the experiment and used as a maximum LDH release control.

### 4.6. Gene Expression Analysis via High-Throughput RT-qPCR

Gene expression analysis was performed as described in [41], using the same gene set. The whole gene set is summarized in Appendix A. Briefly, mRNA isolation was performed on the previously frozen pellets utilizing the NucleoSpin RNA Plus Kit (Macherey-Nagel, Düren, Germany). The resulting mRNA was quantified via absorbance measurement and transcribed into cDNA using the qScript cDNA Synthesis Kit (Quantabio, Beverly, CA, USA). Afterward, cDNA samples underwent a pre-amplification step with a pooled primer mix and TaqMan Preamp Master Mix (Applied Biosystems, Foster City, CA, USA), followed by treatment with Exonuclease I (New England Biolabs (Frankfurt am Main, Germany). The qPCR was performed using a 96 × 96 Dynamic Array integrated fluidic circuit (Fluidigm, San Francisco, CA, USA). The resulting data were analyzed with the Fluidigm Real-Time PCR Analysis and GenEx software. The ΔΔC_q_ method was applied to calculate differences in the resulting C_q_ values of untreated and treated samples. The results were displayed as log_2_-fold changes compared to the untreated control. All gene expression samples were assessed in two replicates.

### 4.7. Interleukin-8 Enzyme-Linked Immunosorbent Assay (IL-8 ELISA)

IL-8 release was assessed with the IL-8 Human Uncoated ELISA Kit (ThermoFisher Scientific, Karlsruhe, Germany) according to the manufacturer’s information. Briefly, 96-wells plates were coated with a capture antibody. Afterward, the plates were blocked, followed by the addition of standard and diluted medium samples from exposure experiments. The plates were incubated for either two hours at room temperature or overnight at 4 °C. A detection antibody and avidin/horseradish peroxidase conjugate were added. Afterward, the plate was incubated with the substrate 3,3′,5,5′-tetramethylbenzidine, and H_2_SO_4_ was added to stop the reaction. A plate reader was used to detect absorption at 450 nm. Positive control cells were treated with 10 µg/mL lipopolysaccharide (LPS) over the whole post-incubation period.

### 4.8. Evaluation of DNA Strand Breaks by Alkaline Unwinding

In contrast to other semi-quantitative methods such as the comet assay, the alkaline unwinding (AU) method allows DSBs to be quantified with high sensitivity and was performed as described previously [108,109]. Briefly, suspensions of 3 × 10^5^ cells in PBS were kept on ice until the AU procedure was performed. All the following steps were performed in the dark. The experiments were regularly conducted in duplicates. Firstly, 1.5 mL of alkaline solution (0.03 M NaOH, 0.02 M Na_2_HPO_4_, 0.9 M NaCl) was added to 20 µL of cell suspension in a glass tube. After an incubation period of 30 min, 0.1 N HCl was added for neutralization to a pH of 6.8. The samples were sonicated for 15 s and 15 µL of sodium dodecylsulfate (10%) was added. The samples were frozen at −20 °C until chromatography was performed. To separate single-stranded and double-stranded DNA, a hydroxylapatite chromatography was carried out at 60 °C. Potassium phosphate buffers at different concentrations were added to elute single-stranded and double-stranded DNA separately. In a final step, the DNA was stained with Hoechst 33,258 and quantified in a plate reader via fluorescence measurement (exc.: 360 nm, em.: 455 nm). The share of double-stranded DNA was calculated as described in a previous work [108].

### 4.9. Statistical Analysis

Statistical analysis was performed by applying a two-tailed Student’s *t*-test to assess the significance of changes between CF-treated and untreated samples.

## 5. Conclusions

In this study, we assessed the toxicological responses to mechanically (mCF) and thermo-mechanically treated PAN-based carbon fibers (tmCF), applied to BEAS-2B mono-, BEAS-2B/dTHP-1 co-, and BEAS-2B/dTHP-1/CCD-33Lu triple-cultures at an air–liquid interface. Although both fiber samples did not induce cytotoxicity, gene expression analysis revealed a moderate inflammatory response upon exposure to mCF, whereas tmCF triggered more pronounced effects such as pro-inflammatory and stress signaling. The release of the cytokine IL-8 corroborated these findings on the protein level. Determination of DNA strand breaks revealed no strong genotoxic effects in the mono- and co-culture. However, the induction of DNA strand breaks in the triple-culture uncovered a possible secondary genotoxicity possibly driven by a prominent pro-inflammatory response. Literature comparison shows several parallels to other carbon-based materials such as CB or MWCNTs. Depending on the share of fiber-shaped fragments in the samples originating from the fiber treatment, CF can be toxicologically placed between these two materials.

The results emphasize the need to further characterize the hazard of CF and the importance of determining exposure levels at workplaces as well as safety measures during the use of CF-containing materials. Moreover, the development and application of more complex in vitro models of the human lung help to establish new approach methodologies in the general field of particle and fiber toxicology and contribute to a reduction in in vivo investigations in the context of the 3R initiative.

## Figures and Tables

**Figure 1 ijms-24-01927-f001:**
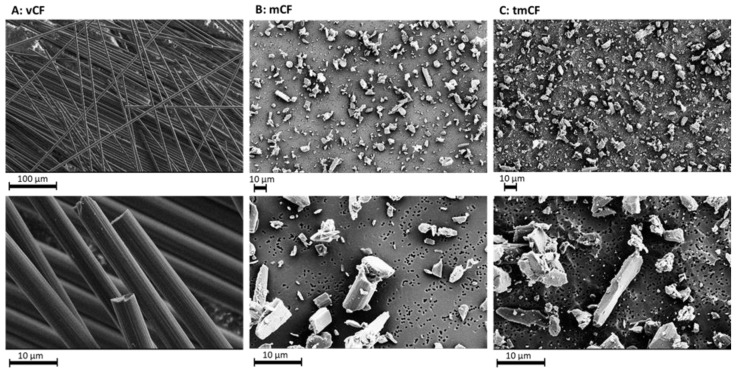
SEM images of the PAN-based CF: (**A**): the original CF as delivered by the manufacturer. Magnification: X200 (top panel), X2000 (lower panel). (**B**): Fiber fragments of mCF deposited on the membrane surface in the air–liquid interface exposure chamber. Magnification: ×500 (top panel), ×2000 (lower panel). (**C**): Fiber fragments of tmCF deposited on the membrane surface in the air–liquid interface exposure chamber. Magnification: ×500 (top panel), ×2000 (lower panel). Images were taken with a Zeiss Supra 55 VP (Zeiss GmbH, Jena, Germany).

**Figure 2 ijms-24-01927-f002:**
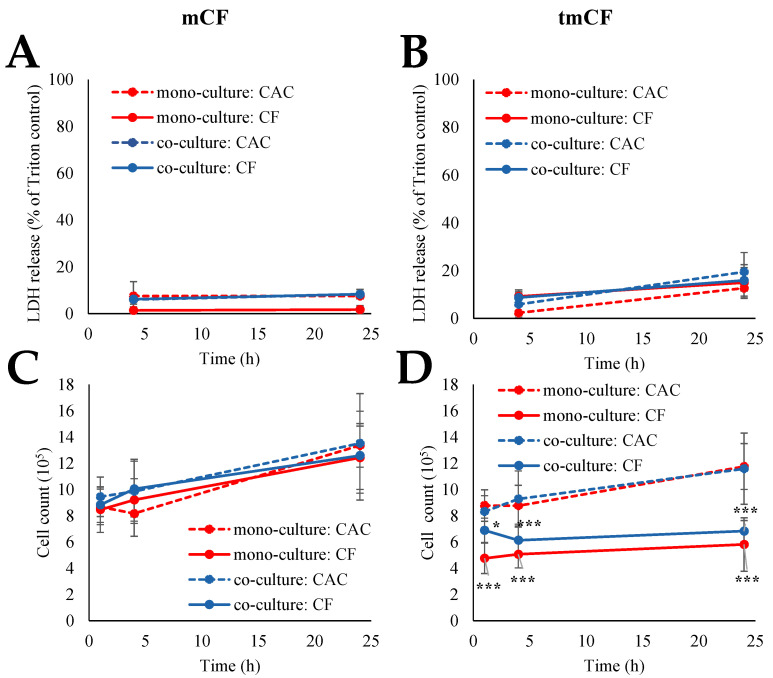
Assessment of cytotoxicity in BEAS-2B mono- and BEAS-2B/dTHP-1 co-cultures after exposure to mechanically treated CF (mCF) (**A**,**C**) and thermo-mechanically treated CF (tmCF) (**B**,**D**). BEAS-2B cells were seeded with a density of 2 × 10^5^/cm^2^. After 4 h, dTHP-1 cells (3.9 × 10^4^/cm^2^) were seeded on top of the BEAS-2B cells for co-culture treatment. On the next day, mono- and co-cultures were exposed to CF samples with average surface doses of 7.00 ± 1.25 µg/cm^2^ (mCF, (**A**,**C**)) or 6.55 ± 1.96 µg/cm^2^ (tmCF, (**B**,**D**)) for 1 h and post-incubated for 0, 3, or 23 h. Clean air controls (CAC) were treated respectively. Cells were harvested for the determination of cell count (**C**,**D**). The supernatants were analyzed for LDH release (**A**,**B**), which is depicted relative to the Triton control (100%). The mean values ± SD of at least three independent experiments performed in duplicates are displayed. Statistical analysis was performed to assess differences between treated and untreated cells using Student’s *t*-test: * (*p* ≤ 0.05), *** (*p* ≤ 0.001).

**Figure 3 ijms-24-01927-f003:**
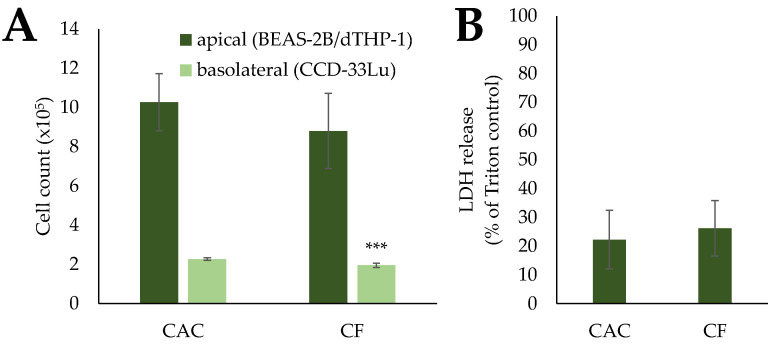
Assessment of cytotoxicity in BEAS-2B/dTHP-1/CCD33-Lu triple-cultures after exposure to mechanically treated CF (mCF). CCD-33Lu cells were seeded on the basolateral side of inverted inserts on the basolateral side of the membrane with a density of 9 × 10^4^/cm^2^. After 1 h, the inserts were inverted, and BEAS-2B cells were seeded onto the apical side of the membrane with a density of 2 × 10^5^/cm^2^. After 4 h, dTHP-1 cells (3.9 × 10^4^/cm^2^) were seeded on top of the BEAS-2B cells. On the next day, triple-cultures were exposed to CF samples with average surface doses of 7.00 ± 1.25 µg/cm^2^ mCF for 1 h and post-incubated for 23 h. Clean air controls (CAC) were treated analogously. Cells were harvested for the determination of cell count (**A**). The supernatants were analyzed for LDH release (**B**), which is depicted relative to the Triton control (100%). The mean values ± SD of at least three independent experiments performed in duplicates are displayed. Statistical analysis was performed to assess differences between treated and untreated cells using Student’s *t*-test: *** (*p* ≤ 0.001).

**Figure 4 ijms-24-01927-f004:**
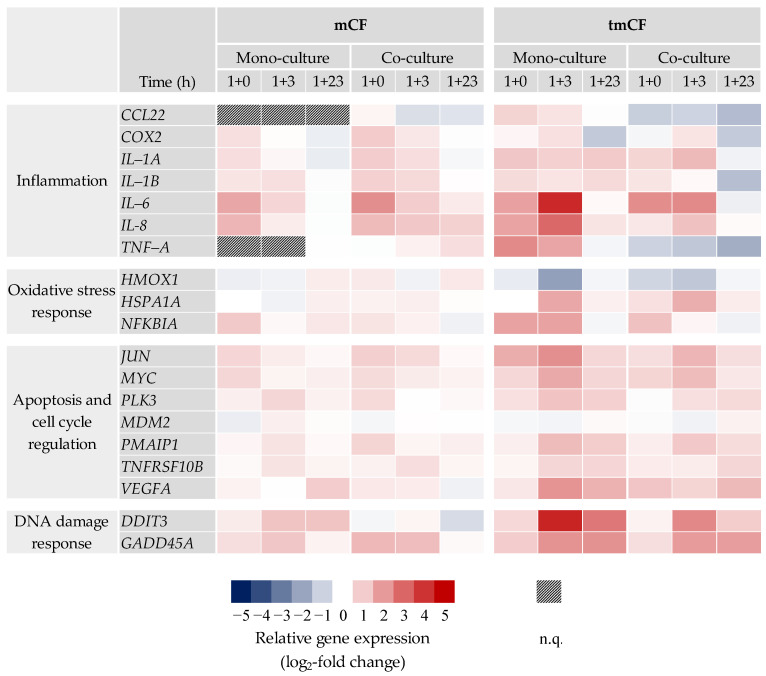
Gene expression profiles of BEAS-2B mono- and BEAS-2B/dTHP-1 co-cultures after exposure to mechanically (mCF) or thermo-mechanically treated CF (tmCF). Both cell culture models were exposed to CF samples with average surface doses of 7.00 ± 1.25 µg/cm^2^ (mCF) or 6.55 ± 1.96 µg/cm^2^ (tmCF) for 1 h and post-incubated for 0, 3, or 23 h. The results are depicted as the log_2_-fold change in the relative gene expression. A red color represents the induction; a blue color, the repression of a gene. The mean values of at least three independent experiments are displayed. n.q.: not quantifiable due to low expression levels.

**Figure 5 ijms-24-01927-f005:**
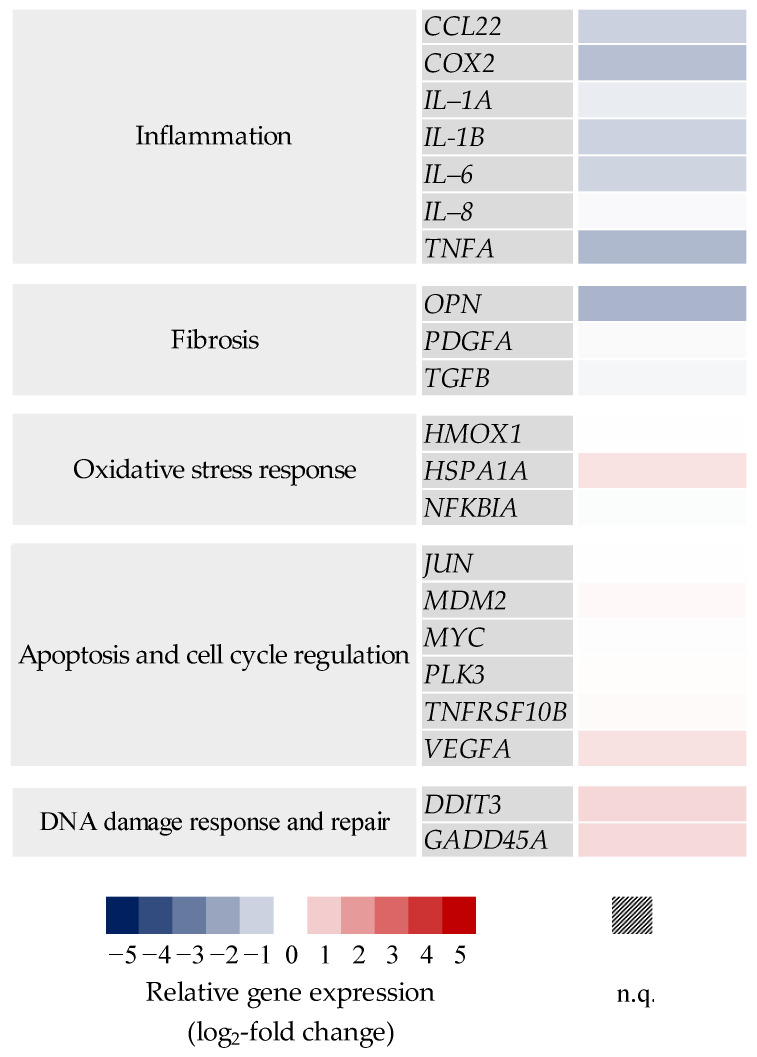
Gene expression profiles of BEAS-2B/dTHP-1/CCD-33Lu triple-cultures after exposure to mechanically treated CF (mCF)**.** Triple-cultures were exposed to CF samples with average surface doses of 7.00 ± 1.25 µg/cm^2^ mCF for 1 h and post-incubated for 23 h. Gene expression changes in cells from the apical compartment (BEAS-2B and dTHP-1) are shown. The results are depicted as the log_2_-fold change in the relative gene expression. A red color represents the induction; a blue color, the repression of a gene. The mean values of at least three independent experiments are displayed. n.q.: not quantifiable due to low expression levels.

**Figure 6 ijms-24-01927-f006:**
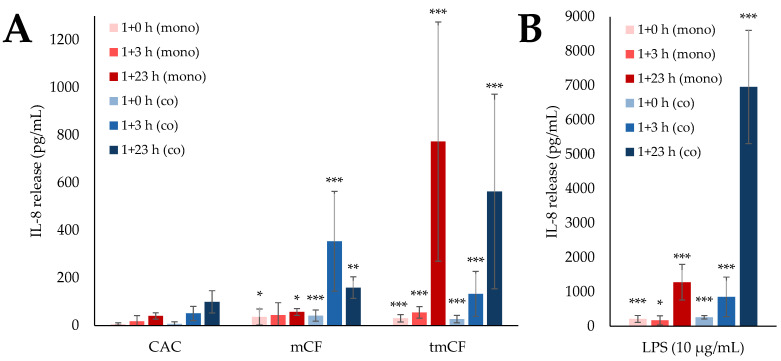
Interleukin-8 (IL-8) release from BEAS-2B mono-cultures (mono, red bars) as well as BEAS-2B/dTHP-1 co-cultures (co, blue bars) after exposure to mechanically (mCF) and thermo-mechanically treated carbon fibers (tmCF) (**A**) and 10 µg/mL lipopolysaccharide (LPS) (**B**). BEAS-2B cells were seeded as mono-cultures at a density of 2 × 10^5^/cm^2^. For co-culture treatment, dTHP-1 cells were seeded on top with a density of 3.95 × 10^4^/cm^2^ after 4 h. The next day, mono- and co-cultures were exposed to CF samples with average surface doses of 7.00 ± 1.25 µg/cm^2^ (mCF) or 6.55 ± 1.96 µg/cm^2^ (tmCF) and post-incubated for 0, 3, or 23 h. Clean air controls (CAC) and LPS exposed cultures received equivalent treatment. The supernatants were analyzed for IL-8 release. The mean values ± SD of three independent experiments performed in duplicates are displayed. Statistical analysis was performed to assess differences between untreated and treated cells using Student’s *t*-test: * (*p* ≤ 0.05), ** (*p* ≤ 0.01), *** (*p* ≤ 0.001).

**Figure 7 ijms-24-01927-f007:**
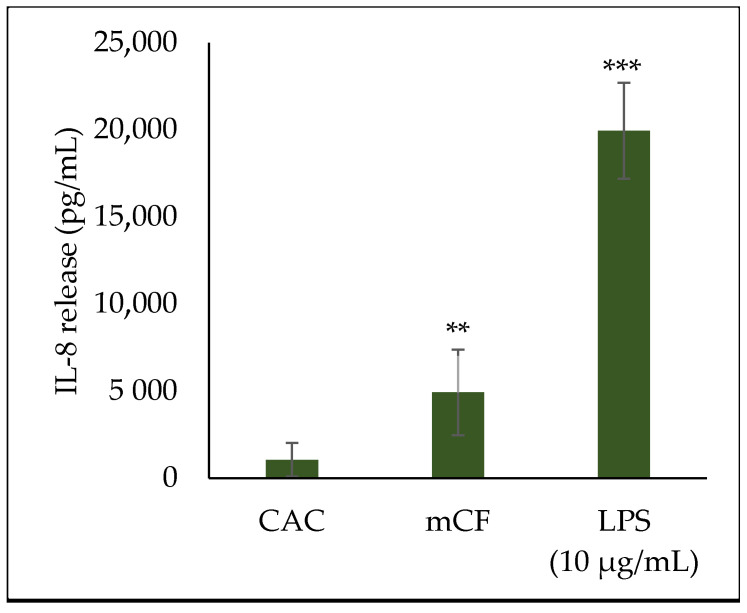
Interleukin-8 (IL-8) release from BEAS-2B/dTHP-1/CCD-33Lu triple-cultures after exposure to mechanically treated carbon fibers (mCF) and 10 µg/mL lipopolysaccharide (LPS). CCD-33Lu cells were seeded on inverted inserts with a density of 9 × 10^4^/cm^2^. After 1 h, the inserts were inverted, and BEAS-2B cells were seeded with a density of 2 × 10^5^/cm^2^. After 4 h, dTHP-1 cells (3.9 × 10^4^/cm^2^) were seeded on top of the BEAS-2B cells. On the next day, mono- and co-culture were exposed to CF samples with average surface doses of 7.00 ± 1.25 µg/cm^2^ mCF for 1 h and post-incubated for 23 h. Clean air controls (CAC) and LPS-exposed cultures received equivalent treatment. The supernatants were analyzed for IL-8 release. The mean values ± SD of at least three independent experiments performed in duplicates are displayed. Statistical analysis was performed to assess differences between untreated and treated cells using Student’s *t*-test: ** (*p* ≤ 0.01), *** (*p* ≤ 0.001).

**Figure 8 ijms-24-01927-f008:**
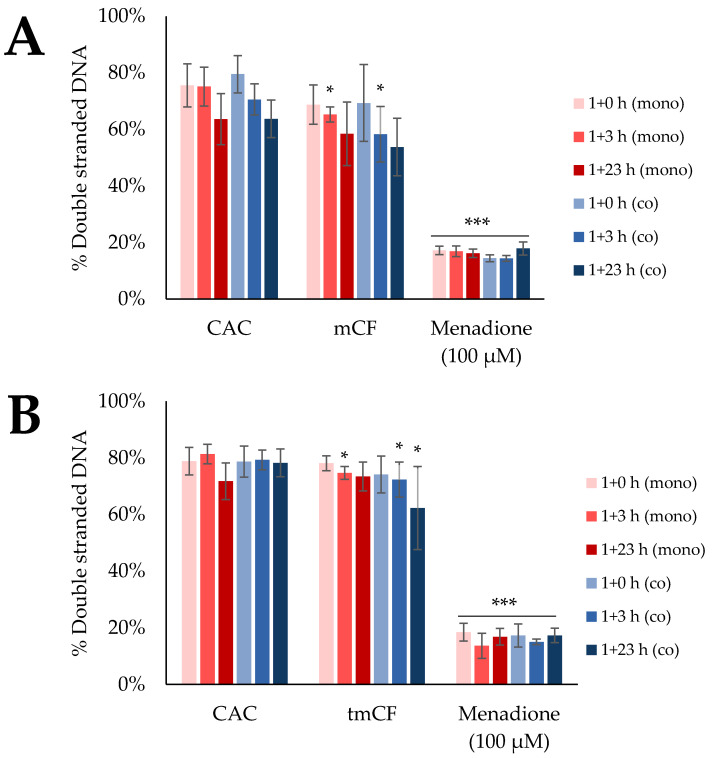
Assessment of DNA strand breaks after exposure of BEAS-2B mono-cultures and BEAS-2B/dTHP-1 co-cultures to mechanically (mCF; (**A**)) or thermo-mechanically treated (tmCF; (**B**)) carbon fibers (CF). BEAS-2B cells were seeded as mono-cultures at a density of 2 × 10^5^/cm^2^. For co-culture treatment, dTHP-1 cells were seeded on top with a density of 3.95 × 10^4^/cm^2^ after 4 h. The next day, mono- and co-cultures were exposed to CF samples with average surface doses of 7.00 ± 1.25 µg/cm^2^ (mCF) or 6.55 ± 1.96 µg/cm^2^ (tmCF) and post-incubated for 0, 3, or 23 h. Clean air controls (CAC) received equivalent treatment. Positive control cultures were exposed to 100 µM menadione 1 h prior to the end of the experiment. The mean values ± SD of three independent experiments performed in duplicates are displayed. Statistical analysis was performed to assess differences between untreated and treated cells using Student’s *t*-test: * (*p* ≤ 0.05), *** (*p* ≤ 0.001).

**Figure 9 ijms-24-01927-f009:**
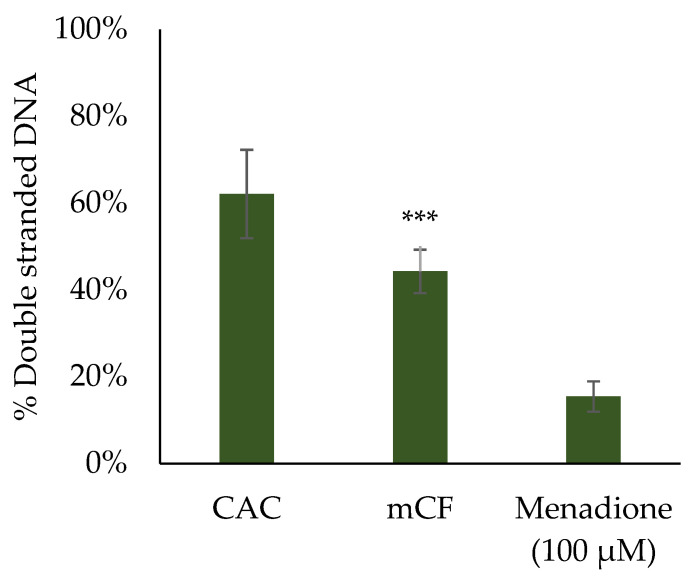
Assessment of DNA strand breaks after exposure of BEAS-2B/dTHP-1/CCD-33Lu triple-cultures to mechanically treated carbon fibers (mCF). CCD-33Lu cells were seeded on inverted inserts with a density of 9 × 10^4^/cm^2^. After 1 h, the inserts were inverted, and BEAS-2B cells were seeded with a density of 2 × 10^5^/cm^2^. After 4 h, dTHP-1 cells (3.9 × 10^4^/cm^2^) were seeded on top of the BEAS-2B cells. On the next day, mono- and co-culture were exposed to an average surface dose of 7.00 ± 1.25 µg/cm^2^ mCF for 1 h and post-incubated for 24 h. Clean air controls (CAC) received equivalent treatment. Positive control cultures were exposed to 100 µM menadione 1 h prior to the end of the experiment. The mean values ± SD of three independent experiments performed in duplicates are displayed. Statistical analysis was performed to assess differences between untreated and treated cells using Student’s *t*-test: *** (*p* ≤ 0.001).

**Table 1 ijms-24-01927-t001:** Exposure and surface doses of mCF and tmCF in the Aerosol Exposure Station.

	mCF	tmCF
**Exposure Dose in AES**		
Number concentration (welas) c_N_ ^a^	2.7 × 10^6^ ± 8.9 × 10^5^ cm^3^	8.8 × 10^6^ ± 3.8 × 10^6^ cm^3^
Aerodynamic equivalent diameter d_ae_ ^a^	3.74 ± 0.89 µm	4.90 ± 0.45 µm
**Relevant in vitro dose on membranes**		
Online mass surface dose (QCM)	7.00 ± 1.25 µg/cm^2^	6.55 ± 1.96 µg/cm^2^
Mass surface dose ^b^	2.46 ± 2.88 µg/cm^2^	4.38 ± 4.58 µg/cm^2^
Number surface dose c_N_ ^b^	4934 ± 2795/cm^2^	3391 ± 3895/cm^2^
Particle fraction	53.7 ± 22.6%	76.5 ± 25.5%
Fiber fraction	26.2 ± 12.1%	14.2 ± 15.1%
WHO fiber fraction	20.1 ± 10.4%	9.4 ± 10.6%

^a^ Aerosol sampled from reactor of AES; ^b^ image evaluation of deposited CF on QCM sensor.

## Data Availability

The data presented in this study are available on request from the first (A.F., S.F.-D.) and corresponding author (A.H., C.W.) for researchers of academic institutes who meet the criteria for access to the confidential data.

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
