# Peer review of "Comparing the Toxicological Responses of Pulmonary Air–Liquid Interface Models upon Exposure to Differentially Treated Carbon Fibers"

_ijms, 2023, doi:10.3390/ijms24031927_

Round 1

Reviewer 1 Report

Does the introduction provide sufficient background and include all relevant references?

Yes, it does. However, the introduction could be improved. The description of the carbon fibers is too long and rich of interesting details respect to the biological research that will be exposed.

Lines 27-30. The period is too long and it is difficult to follow

Line 150-152 The period is too long

Figure 1 The figure is not mentioned in the text . Magnification is not reported.

Line 592 and Line 598. It needs to specify that BEAS-2B and THP-1 cells are from human origin.

Line 631 Could be the “exposure medium” replace with medium necessary for the exposure? It seems more clear.

Reviewer 2 Report

Manuscript "Comparing the Toxicological Responses of Pulmonary Air-Liq- 2 uid Interface Models upon Exposure to Differentially Treated 3 Carbon Fibers" has provided the potential study to establish the toxicological effect of CF and emphasizes the necessity for toxicological screening. 

I have few concern and suggestions for the authors:

Why author has used the Evaluation of DNA Strand Breaks by Alkaline Unwinding, but not by comet? Is there any specific reason?

Does the team also examining the patients samples from the different Hazardous site? Any preclinical study data?

Author should write the Conclusion and limitation of this study. Future perspective in terms of what questions and methods needs to be addressed at global level should be written in the manuscript. 

Author should carefully check the gene names in the results table and the entire manuscript, it should not be Capitalized the whole gene name. Kindly follow the scientific guideline and correct it.   

This study has a good rationale to further conduct the study to explore the genome sequencing to identity the different mutations, that can be exposed by CF. 
